# Associations of Body Mass Index with Ventilation Management and Clinical Outcomes in Invasively Ventilated Patients with ARDS Related to COVID-19—Insights from the PRoVENT-COVID Study

**DOI:** 10.3390/jcm10061176

**Published:** 2021-03-11

**Authors:** Renée Schavemaker, Marcus J. Schultz, Wim K. Lagrand, Eline R. van Slobbe-Bijlsma, Ary Serpa Neto, Frederique Paulus

**Affiliations:** 1Department of Intensive Care, Amsterdam UMC, Location AMC, 1105 AZ Amsterdam, The Netherlands; reneeschavemaker@hotmail.com (R.S.); m.j.schultz@amsterdamumc.nl (M.J.S.); w.k.lagrand@amsterdamumc.nl (W.K.L.); ary.serpaneto@monash.edu (A.S.N.); 2Mahidol Oxford Tropical Medicine Research Unit (MORU), Mahidol University, Bangkok 10400, Thailand; 3Nuffield Department of Medicine, University of Oxford, Oxford OX3 7FZ, UK; 4Department of Intensive Care, Tergooi Hospitals, 1213 XZ Hilversum, The Netherlands; ESlobbeVan-Bijlsma@tergooi.nl; 5Department of Critical Care Medicine, Australian and New Zealand Intensive Care Research Centre (ANZIC-RC), Monash University, Melbourne, VIC 3004, Australia; 6ACHIEVE, Centre of Applied Research, Faculty of Health, Amsterdam University of Applied Sciences, 1105 AZ Amsterdam, The Netherlands

**Keywords:** coronavirus disease 2019, COVID-19, ARDS, body mass index, BMI, normal-weight overweight, obesity, obesity paradox, intensive care, critical care, artificial ventilation, mortality

## Abstract

We describe the practice of ventilation and mortality rates in invasively ventilated normal-weight (18.5 ≤ BMI ≤ 24.9 kg/m^2^), overweight (25.0 ≤ BMI ≤ 29.9 kg/m^2^), and obese (BMI > 30 kg/m^2^) COVID-19 ARDS patients in a national, multicenter observational study, performed at 22 intensive care units in the Netherlands. The primary outcome was a combination of ventilation variables and parameters over the first four calendar days of ventilation, including tidal volume, positive end–expiratory pressure (PEEP), respiratory system compliance, and driving pressure in normal–weight, overweight, and obese patients. Secondary outcomes included the use of adjunctive treatments for refractory hypoxaemia and mortality rates. Between 1 March 2020 and 1 June 2020, 1122 patients were included in the study: 244 (21.3%) normal-weight patients, 531 (47.3%) overweight patients, and 324 (28.8%) obese patients. Most patients received a tidal volume < 8 mL/kg PBW; only on the first day was the tidal volume higher in obese patients. PEEP and driving pressure were higher, and compliance of the respiratory system was lower in obese patients on all four days. Adjunctive therapies for refractory hypoxemia were used equally in the three BMI groups. Adjusted mortality rates were not different between BMI categories. The findings of this study suggest that lung-protective ventilation with a lower tidal volume and prone positioning is similarly feasible in normal-weight, overweight, and obese patients with ARDS related to COVID-19. A patient’s BMI should not be used in decisions to forgo or proceed with invasive ventilation.

## 1. Introduction

Several studies have implicated obesity as a risk factor for complications in COVID-19, such as the development of severe pneumonia and acute respiratory failure [1,2], the need for hospitalization [3], the need for intensive care unit (ICU) admission [3,4], and the need for invasive ventilation [5]. Multiple pathways by which obesity may affect outcomes in COVID-19 patients have been suggested, including underlying impairments in respiratory, cardiovascular, metabolic, and thrombotic pathways [1]. It is uncertain, though, whether obesity should be considered in the decision of who undergoes or will continue with treatment, including intubation and invasive ventilation.

Invasive ventilation can save the life of patients with acute respiratory distress syndrome (ARDS), but can also cause harm if not properly applied. Ventilatory interventions, such as lung-protective ventilation with a lower tidal volume [6] or a lower driving pressure [7], appropriate titration of positive end-expiratory pressure (PEEP) with recruitment maneuvers [8,9], and prone positioning [10] all have the potential to improve outcome of invasively ventilated patients. Obesity may hamper adequate use of these interventions and may increase the risk of severe atelectasis, but at the same time redistribute regional transpulmonary pressure, possibly reducing the potential negative effects of invasive ventilation in an inhomogeneous lung [11]. It is not clear how body mass index (BMI) affects ventilation practice and outcomes in patients with ARDS related to COVID-19.

In this secondary analysis of the “PRactice of Ventilation in COVID-19” (PRoVENT-COVID) study [12,13], we aimed to describe and compare ventilation management over the first 4 days of invasive ventilation in normal-weight, overweight and obese COVID-19 patients with ARDS. One secondary objective was to describe and compare clinical outcomes in the three BMI categories. The hypotheses tested were that ventilation practices differ between normal-weight, overweight, and obese patients with ARDS related to COVID-19, and that obese patients have a worse outcome compared to normal-weight and overweight patients.

## 2. Materials and Methods

### 2.1. Study Design

The PRoVENT-COVID study is an investigator-initiated, multicenter, observational cohort study undertaken at 22 ICUs during the first 3 months of the COVID-19 outbreak in the Netherlands [13]. The study protocol including a statistical analysis plan was prepublished [12]. A statistical analysis plan for the current analysis was published online before assessing the database of the PRoVENT-COVID study [14]. Study sites were recruited through direct contact by steering committee members of PRoVENT-COVID. Study coordinators contacted the local doctors, trained data collectors to assist local caregivers, and monitored the study according to the International Conference on Harmonization Good Clinical Practice Guidelines. The integrity and timely completion of data collection was ensured by the study coordinators.

### 2.2. Ethics

Ethical approval for this study (W20_157 # 20.171) was provided by the Ethical Committee of the Academic Medical Center (Chairperson C.L. van der Wilt) on 7 April 2020. The need for individual patient informed consent was waived given the observational design of the study.

### 2.3. Study Registration

The study was registered at clinicaltrials.gov (15 April 2020; study identifier NCT04346342).

### 2.4. Inclusion and Exclusion Criteria

The PRoVENT-COVID study had the following inclusion criteria: (1) age ≥ 18 years, (2) admitted to one of the participating ICUs, and (3) having received invasive ventilation for ARDS related to COVID-19. The study itself had no exclusion criteria; for the current analysis we excluded patients in whom BMI could not be calculated.

### 2.5. Collected Data, and Patient Classification

Demographics and data regarding premorbid diseases and medication were collected at baseline. In the first hour of invasive ventilation and every 8 h thereafter, at fixed time points in the first four calendar days, ventilator settings and parameters were collected. The driving pressure and mechanical power of ventilation were calculated as follows: driving pressure (in cm H_2_O) = peak pressure − positive end-expiratory pressure (PEEP); and mechanical power (in J/min) = 0.098 × tidal volume × respiratory rate × (peak pressure-0.5 × driving pressure). The categories of BMI were defined as underweight (BMI < 18.4 kg/m^2^), normal-weight (18.5 ≤ BMI ≤ 24.9 kg/m^2^), overweight (25.0 ≤ BMI ≤ 29.9 kg/m^2^), and obese (BMI > 30 kg/m^2^).

### 2.6. Endpoints

The primary outcome of this analysis was a combination of key ventilator settings and ventilation parameters during the first four calendar days of invasive ventilation, including tidal volume, PEEP, respiratory system compliance, and driving pressure. Secondary outcomes included the use of adjunctive treatments for refractory hypoxemia, including the use of alveolar recruitment maneuvers and prone positioning. We also collected adjunctive strategies including the use of neuromuscular blocking agents and extracorporeal membrane oxygenation. Other secondary outcomes included mortality at day 28, at ICU and hospital discharge, and at day 90, ventilator-free days and alive at day 28 (VFD-28) as defined before [15], and typical complications including venous thromboembolism, acute kidney injury, and the use of renal replacement therapy.

### 2.7. Statistical Analysis

Since the proportion of patients with underweight was very low (*n* = 3), underweight patients were excluded from the analyses. No statistical power calculation was conducted before the study, and sample size was based on available data. The amount of missing data was low for most of the variables and follow-up to day 90 was complete for 91% of patients.

Continuous variables are presented as medians (first quartile–third quartile) and categorical variables as numbers and percentages. The BMI groups were compared using Kruskal–Wallis test for continuous variables and Fisher exact tests for categorical variables.

Differences in ventilatory variables and laboratory tests between the BMI groups are visualized in cumulative distribution plots and boxplots at the start of ventilation and at day 1, 2, and 3. The effect of BMI categories on clinical outcomes was reported in Kaplan–Meier curves and compared with Log-rank tests. Further comparison between the groups was made with (shared-frailty) Cox proportional models with center as frailty. The proportional hazard assumption was assessed through Schoenfeld residuals.

To assess the impact of BMI categories on 28-day mortality, the following variables were included in the multivariable model based on clinical relevance and when a *p* < 0.20 was found in the univariable assessment: (1) ventilatory variables in the first day aggregated as the mean from a maximum of four assessments (tidal volume adjusted by predicted body weight (PBW) and respiratory system compliance and PEEP); (2) laboratory tests and vital signs on the first day aggregated as the mean from a maximum of four assessments (arterial pH, creatinine, heart rate, and mean arterial pressure); (3) organ support at the first day (use of vasopressor and cumulative fluid balance); (4) demographic characteristics (age, gender, body mass index, hypertension, heart failure, diabetes, chronic kidney disease, chronic obstructive pulmonary disease, active hematological neoplasia, active solid neoplasia, use of angiotensin converting enzyme inhibitor, and use of angiotensin II receptor blocker); (5) severity of ARDS according to the Berlin definition [16]; and (6) use of rescue therapies at the first day of ventilation. Peak pressure and driving pressure were not considered due to collinearity with respiratory compliance, which was judged to be more clinically relevant in the model, and FiO2 was excluded due to association with PaO2/FiO2 and the severity of acute respiratory failure.

When predictors considered in the model were missing in less than 5% of the patients, these values were imputed by the median. All continuous variables were entered after standardization to improve the convergence of the model, and all effect estimates represent the increase in one standard deviation of the variable.

Two sensitivity analyses were performed. First, the impact of BMI categories on 28-day mortality was re-assessed within the mild, moderate, and severe categories of ARDS according to the Berlin definition [16]. In addition, obese categories class I (30.0 ≤ BMI ≤ 34.9 kg/m^2^), II (35.0 kg/m^2^ ≤ BMI ≤ 39.9 kg/m^2^), and III (BMI ≥ 40 kg/m^2^) were considered and assessed in this model.

All analyses were conducted in R v.4.0.2 (R Foundation, Vienna, Austria) and the significance level was set at 0.05.

## 3. Results

### 3.1. Participating ICUs and Patients Enrolled

Between 1 March 2020 through 1 June 2020, 31 ICUs were invited for participation in PRoVENT-COVID, and 22 met inclusion criteria. Of 1340 individuals screened, 1122 were enrolled; the main reasons for exclusion were that they did not receive invasive ventilation or had an alternative diagnosis (Figure 1). Three (0.2%) patients were under-weight, 244 (21.3%) patients were normal-weight, 531 (47.3%) patients were overweight, and 324 (28.8%) patients were obese; in 20 patients BMI could not be calculated. Obese patients were younger, less often males, had more often a history of diabetes and less often chronic kidney injury (Table 1). Obese patients more often presented with severe ARDS, had a higher mean arterial pressure, and required less often inotropic agents on the first day of ventilation.

### 3.2. Ventilatory Support and Adjunctive Therapies

At the day of start of invasive ventilation, obese patients received ventilation with a slightly higher median tidal volume (6.4 (5.9–6.9) vs. 6.4 (5.9–7.0) vs. 6.6 (5.9–7.5) mL/kg PBW in normal-weight vs. overweight vs. obese patients (*p* < 0.001)) (Table 1 and Figure 2), a higher PEEP (12.0 (10.0–14.0) vs. 12.7 (11.0–14.5) vs. 14.0 (12.0–15.0) cm H_2_O (*p* < 0.001)), and a higher driving pressure (13.0 (11.2–15.3) vs. 13.7 (12.0–16.0) vs. 14.5 (12.5–17.0) cm H_2_O (*p* < 0.001)). Obese patients had a lower compliance of the respiratory system (36.2 (28.7–45.1) vs. 33.4 (26.8–41.1) vs. 31.9 (26.0–38.1) ml/cm H_2_O (*p* < 0.001)). Obese patients were ventilated with higher mechanical power and needed higher oxygen fractions. The four ventilatory variables did not change at successive days (Appendix A), and the difference in tidal volume between the three BMI categories was no longer present at day 2, 3, and 4.

In the first 4 days of ventilation, the use of alveolar recruitment maneuvers and prone positioning was not statistically different between the BMI categories (Table 2). Neuromuscular blocking agents were more often administered in obese patients.

### 3.3. Patient Outcomes

The 28-day mortality was similar in the three BMI categories (29.8% vs. 30.9% vs. 23.9% in normal-weight vs. overweight vs. obese patients (*p* = 0.082 for the Fisher exact test; *p* = 0.090 for the Log-rank test) (Figure 3). BMI category was not associated with 28-day mortality, neither in the unadjusted nor in the adjusted analyses (Table 3, Appendix A). Among secondary outcomes, ICU, hospital, and 90-day mortality were lower in obese patients in the unadjusted analysis (Table 2 and Appendix A), but not in the adjusted analysis (Table 3).

In the unadjusted analysis, 28-day mortality was lower in obese patients with moderate ARDS (*p* = 0.040 for the Log-Rank test) (Figure 3). However, there was no effect of BMI categories within each category of ARDS after multivariable adjustment in 28-day, ICU, hospital, or 90-day mortality (Appendix A).

Baseline characteristics of the patients according to five BMI categories are shown in Appendix A. On the first day of ventilation, tidal volume, PEEP, and driving pressure were higher, and compliance was lower, in more obese patients (Appendix A). The 28-day mortality was lower in patients with obesity class III in the unadjusted analysis (Appendix A). After adjustment for confounders, only the higher risk of 28-day mortality in patients with class II obesity and severe ARDS remained statistically significant (Appendix A).

## 4. Discussion

We here describe the associations of BMI with ventilation parameters and outcomes in COVID-19 patients with ARDS who received invasive ventilation during the first 3 months of the outbreak in the Netherlands. First, while tidal volume was slightly higher in obese patients compared to overweight and normal-weight patients at the day of start of ventilation, still >70% of them received protective ventilation with a low tidal volume. Of note, the tidal volume differences were small and probably clinically irrelevant. The tidal volumes became equally large in the three BMI categories on successive days. Second, the applied PEEP and driving pressure were higher and compliance was lower in obese patients. These differences persisted over the successive days. Off all adjunctive treatments for refractory hypoxemia, only neuromuscular blocking agents were more often used in obese patients. Third, 28-day mortality was not different between the BMI groups, and although ICU, hospital, and 90-day mortality were lower in obese patients in the unadjusted analysis, but this was no longer the case in the adjusted analysis. While survival seemed better in obese patients with mild ARDS, this finding disappeared after adjustments. It is possible that mild ARDS was relatively over-diagnosed due to derecruitment, causing oxygenation impairments rather than that a patient really having ARDS.

Previous studies showed that obese patients often receive ventilation with a higher tidal volume [17,18,19] and higher PEEP [18,19], probably because clinicians frequently overestimate lung size by using actual body weight instead of PBW and because obese patients are more susceptible to atelectasis. Obese patients also receive ventilation with a higher plateau and peak pressures [18,19], probably to compensate for the decreased lung and chest wall compliance. In addition, obese patients frequently need a higher FiO2 and a higher minute volume because they consume more oxygen and produce more carbon dioxide [20,21]. We found similar differences between the BMI categories in our cohort of COVID-19 patients, but the effects of BMI on tidal volume were much less outspoken and disappeared over the successive days. One explanation for the latter finding could be that compliance with existing guidelines is much better in COVID-19 patients, either because care for the surge in COVID-19 patients had to be provided by hospital personnel who had less experience or confidence with setting a ventilator and thus followed the local guidelines more strictly, or because the use of ventilation with a lower tidal volume has improved in general over recent years. Nevertheless, the use of a correct tidal volume is important also in COVID-19 patients with ARDS [13].

Esophagus catheters were seldom used in this cohort of patients, probably because hospital personnel with little experience with setting a ventilator also had less knowledge of transpulmonary and transthoracic pressure measurements to adjust ventilator settings. It remains uncertain whether (adequate) use of esophagus pressure measurements would have altered the current findings [22].

Use of prone positioning was high, but not different between BMI categories. This was surprising as prone positioning in an obese patient could be more burdensome than in overweight or normal-weight patients. It is probable that the common use of the prone positioning is due to the high incidence of severe hypoxemia in COVID-19 patients with ARDS. Prone positioning has been indisputably shown to improve oxygenation in patients with ARDS [23]. Neuromuscular blocking agents were prescribed more often in obese patients. Obese patients may have more dyspnea because of a higher minute volume needed to compensate for the greater production of carbon dioxide. This may increase patient-ventilator asynchronies, for which clinicians could prescribe neuromuscular blocking agents.

The effects of BMI on mortality in the current study contrast with the findings of one meta-analysis showing a significant association between obesity and COVID-19 severity and outcome [24]. That meta-analysis, however, accepted studies that included patients other than those receiving invasive ventilation in the ICU; the impact of BMI on outcome could differ between ventilated and non-ventilated COVID-19 patients. We cannot exclude the possibility that decisions to forgo invasive ventilation were driven by factors that may impact outcome, including the presence of comorbidities and the health status before development of COVID-19 pneumonia. This may have resulted in BMI categories with comparable mortality rates. Nevertheless, in our cohort obese patients had diabetes more often. The incidence of chronic kidney injury, however, was lower in obese patients.

Our findings are also different from the results of two meta-analyses showing better outcomes in invasively ventilated obese patients compared to overweight or normal-weight patients [25,26]. In the meta-analysis of studies of invasively ventilated ICU patients [25], obese patients had a lower ICU, hospital, and long-term mortality. In the meta-analysis of studies of invasively ventilated ARDS patients [26], obesity and morbid obesity were more likely to result in a lower mortality. These meta-analyses, however, lacked adjustments for age, gender, comorbidities, and illness severity. In our cohort, differences in outcomes between BMI categories disappeared in the adjusted analyses [11].

One important limitation of the current analysis was that we could not use esophageal pressures to calculate pulmonary compliance; indeed, we could only report total respiratory system compliance. In an obese patient, an elevated plateau pressure may be related to an elevated transthoracic pressure and not necessarily an increase in transpulmonary pressure. This may explain, at least in part, the finding that there is no significant difference in mortality in severe ARDS (between obese and normal-weight) and even the important difference in favor of obese patients in mild and moderate ARDS. It may also explain the finding that obese patients received ventilation with a higher driving pressure and higher mechanical power and had a lower respiratory system compliance [22].

This analysis has other limitations, some of which have been mentioned before [13]. First, the collection of ventilation variables and adjunctive treatments was restricted to the first 4 days of ventilation, and we cannot exclude the possibility that ventilation practices and use of adjunctive treatments beyond the first days of invasive ventilation also had an effect on outcome. Follow-up to day 90 was not complete for all patients, but we missed follow-up at day 28 for only 30 patients, mainly because these patients were transferred to a non-participating hospital. Third, the models were not adjusted for laboratory test results such as D-dimers, which were not measured daily as part of standard care and were therefore not collected. However, patients with COVID-19 ARDS with increased D-dimer concentrations have higher mortality rates [27]. Nevertheless, one recent study showed that obese COVID-19 patients have lower D-dimer concentrations than non-obese COVID-19 patients [28].

The findings of this analysis extend our knowledge of ventilation practice in normal–weight, overweight, and obese patients with ARDS in general and in COVID-19 patients with ARDS in particular. Furthermore, they provide important information about the outcomes of invasively ventilated patients in the three BMI categories. The design of the study assured the completeness of data collection. The short timeframe within which data were gathered, only 3 months, avoided the effect of practice changes over time.

Our findings may have important suggestions for the clinical management of obese COVID-19 patients with ARDS. Despite understandable differences in ventilation management, it was noticeable that providing lung-protective ventilation with a lower tidal volume and prone positioning are very feasible strategies in obese patients with ARDS related to COVID-19. Both strategies have been proven to be very effective in patients with ARDS [6,10] and have been advised in obese patients with ARDS from another origin [29]; we recently showed that tidal volume size has an independent effect on outcome in COVID-19 patients with ARDS [13]. Most importantly, our findings suggest that the patient’s BMI should not be used in decisions to forgo or proceed with invasive ventilation in patients with ARDS related to COVID-19.

## 5. Conclusions

Comparable to normal-weight, overweight, and obese patients with ARDS from another origin, lung-protective ventilation with a lower tidal volume and prone positioning is very feasible in normal-weight, overweight, and obese patients with ARDS related to COVID-19. Obese patients with ARDS related to COVID-19 do not have a worse outcome compared to normal-weight and overweight patients; therefore, obesity should not be considered in the decision of who undergoes or will continue with intubation and invasive ventilation.

## Figures and Tables

**Figure 1 jcm-10-01176-f001:**
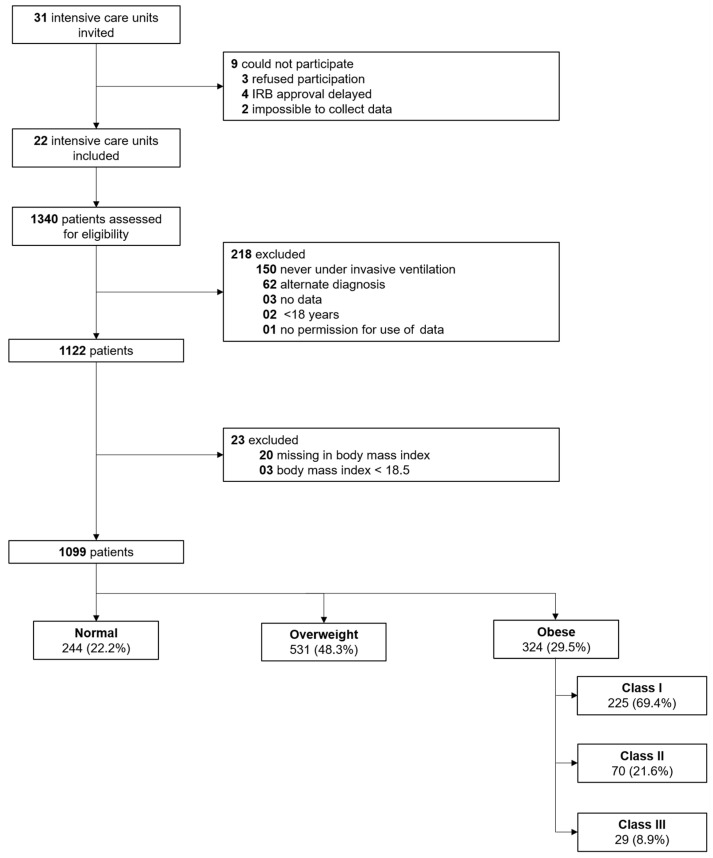
Study profile. Follow-up to 90 days was complete in 996 patients. IRB: Institutional Review Board.

**Figure 2 jcm-10-01176-f002:**
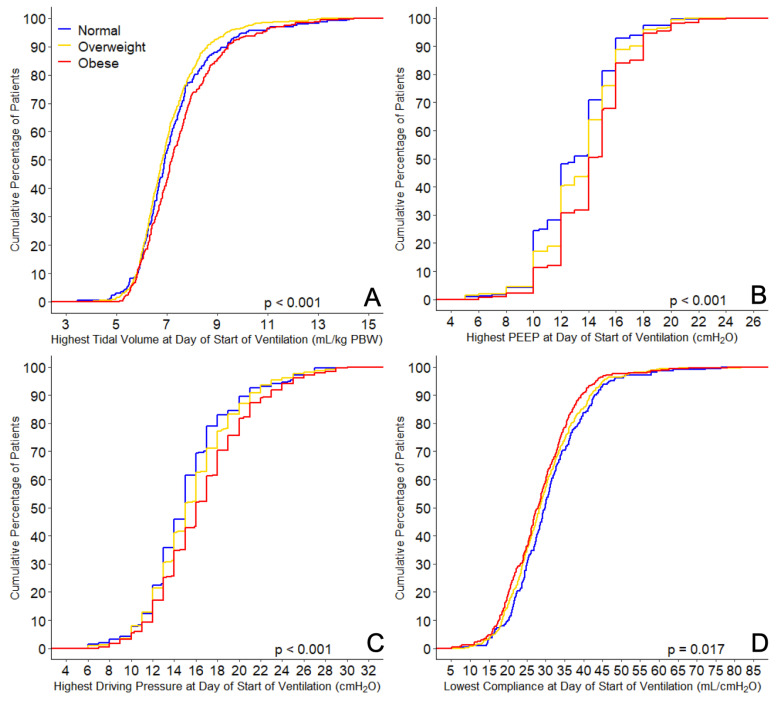
Ventilation parameters. Cumulative frequency distribution of (**A**) tidal volume, (**B**) positive end-expiratory pressure (PEEP), (**C**) driving pressure, and (**D**) respiratory system compliance. *p* values calculated from Kruskal–Wallis tests.

**Figure 3 jcm-10-01176-f003:**
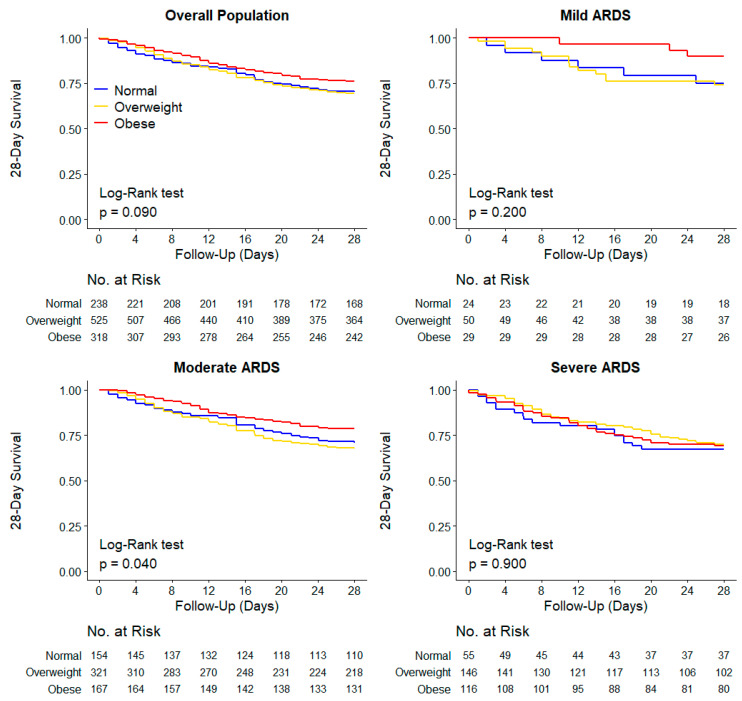
Kaplan–Meier Curves for 28-day mortality in the overall population and groups according to ARDS severity. *p* values calculated from Log-rank test. Unadjusted and adjusted (shared-frailty) Cox proportional hazard models are shown in Appendix A.

**Table 1 jcm-10-01176-t001:** Baseline characteristics, vital signs, laboratory test results, and organ support on the first day of ventilation according to BMI category.

	Normal (*n* = 244)	Overweight (*n* = 531)	Obese (*n* = 324)	*p* Value
Age, years	67.0 (60.0–73.0)	66.0 (59.0–73.0)	61.0 (53.0–70.0)	<0.001
Male gender—no (%)	186 (76.2)	407 (76.6)	209 (64.5)	<0.001
Body mass index, kg/m^2^	23.9 (22.9–24.6)	27.3 (26.2–28.5)	32.9 (31.2–35.9)	<0.001
Transferred under invasive ventilation	51 (20.9)	82 (15.4)	59 (18.2)	0.160
Days between intubation and admission	0.0 (0.0–0.0)	0.0 (0.0–0.0)	0.0 (0.0–0.0)	0.583
Use of non-invasive ventilation—no (%)	15 (6.8)	45 (9.4)	24 (8.1)	0.517
Duration of non-invasive ventilation, hours	5.5 (2.0–48.0)	8.0 (2.0–15.1)	7.5 (2.0–14.8)	0.803
Chest CT scan performed—no (%)	83/232 (35.8)	169/509 (33.2)	103/314 (32.8)	0.738
Lung parenchyma affected—no (%)				0.839
0%	3/84 (3.6)	6/172 (3.5)	5/103 (4.9)	
25%	32/84 (38.1)	51/172 (29.7)	33/103 (32.0)	
50%	26/84 (31.0)	52/172 (30.2)	28/103 (27.2)	
75%	20/84 (23.8)	53/172 (30.8)	29/103 (28.2)	
100%	3/84 (3.6)	10/172 (5.8)	8/103 (7.8)	
Chest X-ray performed—no (%)	127/149 (85.2)	289/334 (86.5)	175/204 (85.8)	0.907
Quadrants affected—no (%)				0.068
1	14/126 (11.1)	21/291 (7.2)	7/173 (4.0)	
2	24/126 (19.0)	67/291 (23.0)	48/173 (27.7)	
3	37/126 (29.4)	72/291 (24.7)	55/173 (31.8)	
4	51/126 (40.5)	131/291 (45.0)	63/173 (36.4)	
Severity of ARDS—no (%)				0.028
Mild	24/239 (10.0)	51/523 (9.7)	29/318 (9.1)	
Moderate	157/239 (65.7)	326/523 (62.3)	171/318 (53.8)	
Severe	58/239 (24.3)	146/523 (27.9)	118/318 (37.1)	
Co-existing disorders—no (%)				
Hypertension	74 (30.3)	186 (35.0)	114 (35.2)	0.383
Heart failure	9 (3.7)	31 (5.8)	8 (2.5)	0.058
Diabetes	41 (16.8)	115 (21.7)	90 (27.8)	0.007
Chronic kidney disease	8 (3.3)	31 (5.8)	8 (2.5)	0.045
Baseline creatinine, µmol/L *	77.0 (61.0–98.0)	78.0 (64.0–97.0)	76.0 (62.8–97.0)	0.767
Liver cirrhosis	0 (0.0)	2 (0.4)	1 (0.3)	0.999
Chronic obstructive pulmonary disease	20 (8.2)	41 (7.7)	24 (7.4)	0.934
Active hematological neoplasia	5 (2.0)	10 (1.9)	1 (0.3)	0.091
Active solid neoplasia	6 (2.5)	14 (2.6)	7 (2.2)	0.967
Neuromuscular disease	2 (0.8)	3 (0.6)	3 (0.9)	0.728
Immunosuppression	9 (3.7)	8 (1.5)	7 (2.2)	0.165
Previous medication—no (%)				
Systemic steroids	10 (4.1)	17 (3.2)	11 (3.4)	0.786
Inhalation steroids	21 (8.6)	58 (10.9)	45 (13.9)	0.137
Angiotensin converting enzyme inhibitor	33 (13.5)	93 (17.5)	60 (18.5)	0.250
Angiotensin II receptor blocker	24 (9.8)	57 (10.7)	44 (13.6)	0.318
Beta-blockers	40 (16.4)	98 (18.5)	71 (21.9)	0.235
Insulin	14 (5.7)	38 (7.2)	26 (8.0)	0.590
Metformin	29 (11.9)	77 (14.5)	65 (20.1)	0.021
Statins	70 (28.7)	155 (29.2)	100 (30.9)	0.826
Calcium channel blockers	47 (19.3)	79 (14.9)	67 (20.7)	0.067
Vital signs at the day of start of ventilation				
Heart rate, bpm **	84.0 (71.5–97.0)	84.0 (73.0–97.1)	86.0 (76.9–98.0)	0.130
Mean arterial pressure, mmHg **	78.7 (73.0–86.0)	80.0 (73.5–87.5)	82.0 (75.7–89.5)	0.002
Laboratory tests at the day of start of ventilation				
pH **	7.36 (7.30–7.41)	7.37 (7.32–7.41)	7.36 (7.31–7.41)	0.700
Worst PaO_2_/FiO_2_, mmHg ***	130.0 (101.0–166.9)	125.0 (98.5–162.4)	114.6 (87.6–146.0)	0.001
PaCO_2_, mmHg **	44.5 (39.5–51.3)	44.3 (38.8–49.6)	44.6 (39.8–51.0)	0.356
Lactate mmol/L **	1.2 (0.9–1.5)	1.2 (1.0–1.4)	1.1 (0.9–1.4)	0.132
Organ support at the day of start of ventilation—no (%)				
Continuous sedation	231/243 (95.1)	506/530 (95.5)	316 (97.5)	0.213
Inotropic or vasopressor	192/243 (79.0)	412/530 (77.7)	241 (74.4)	0.377
Vasopressor	192/243 (79.0)	412/530 (77.7)	240 (74.1)	0.328
Inotropic	18/243 (7.4)	17/530 (3.2)	10 (3.1)	0.022
Fluid balance, mL	696.7 (29.0–1441.0)	515.8 (7.3–1239.3)	449.0 (-15.0–1299.9)	0.171
Urine output, mL	692.5 (333.8–1116.2)	647.5 (350.0–1145.0)	705.0 (395.0–1115.0)	0.500
Ventilation support at the day of start of ventilation				
Assisted ventilation—no (%)	73/243 (30.0)	161/527 (30.6)	84 (25.9)	0.328
Tidal volume, mL/kg PBW **	6.4 (5.9–6.9)	6.4 (5.9–7.0)	6.6 (5.9–7.5)	< 0.001
PEEP, cmH_2_O **	12.0 (10.0–14.0)	12.7 (11.0–14.5)	14.0 (12.0–15.0)	< 0.001
Peak pressure, cmH_2_O **	25.2 (22.8–28.9)	26.6 (23.5–29.3)	28.0 (25.3–31.0)	< 0.001
Driving pressure, cmH_2_O **	13.0 (11.2–15.3)	13.7 (12.0–16.0)	14.5 (12.5–17.0)	< 0.001
Mechanical power, J/min **	18.1 (14.7–21.6)	18.2 (15.3–21.9)	19.4 (15.8–23.5)	0.014
Compliance, mL/cmH_2_O **	36.2 (28.7–45.1)	33.4 (26.8–41.1)	31.9 (26.0–38.1)	< 0.001
Total respiratory rate, mpm **	21.7 (19.3–24.0)	21.7 (19.8–24.0)	22.0 (19.2–24.0)	0.921
Minute ventilation, L/min	9.8 (8.5–11.4)	9.6 (8.4–11.2)	9.7 (8.3–11.2)	0.556
Minute ventilation corrected, mL/kg/min PBW	137.2 (122.3–157.5)	138.3 (122.4–157.3)	141.3 (125.7–164.1)	0.053
FiO_2_ **	0.54 (0.45–0.65)	0.57 (0.47–0.66)	0.60 (0.52–0.71)	<0.001
etCO_2_, mmHg **	35.7 (32.0–40.7)	36.5 (32.4–41.6)	38.4 (34.6–43.8)	<0.001
Rescue therapy at the day of start of ventilation—no (%)				
Prone positioning	61/241 (25.3)	161/522 (30.8)	104/317 (32.8)	0.142
Duration, hours	9.0 (6.0–14.0)	8.0 (4.0–13.5)	8.0 (3.1–13.0)	0.138
Recruitment maneuver	3/197 (1.5)	12/434 (2.8)	5/268 (1.9)	0.641
ECMO	1/241 (0.4)	0/523 (0.0)	3/318 (0.9)	0.066
Use of NMBA	54/243 (22.2)	154/529 (29.1)	89 (27.5)	0.128
Duration, hours	0.0 (0.0–0.0)	0.0 (0.0–8.0)	0.0 (0.0–8.0)	0.182

Data are median (first quartile–third quartile) or No (%). Percentages may not total 100 because of rounding. CT: computed tomography; PEEP: positive end expiratory pressure; ECMO: extracorporeal membrane oxygenation; FiO_2_: inspired fraction of oxygen; PEEP: positive end-expiratory pressure; NMBA: neuromuscular blocking agent. * Most recent measurement in 24 h before intubation or at ICU admission under invasive ventilation. ** Aggregate as the mean of all values available at the first day of ventilation. *** Worst value of four available.

**Table 2 jcm-10-01176-t002:** Clinical outcomes according to BMI category.

	Normal (*n* = 244)	Overweight (*n* = 531)	Obese (*n* = 324)	*p * Value
28-day mortality—no. (%)	71/238 (29.8)	162/525 (30.9)	76/318 (23.9)	0.082
Ventilator-free days at day 28, days	2.0 (0.0–18.0)	0.0 (0.0–15.0)	6.0 (0.0–17.0)	0.088
Duration of ventilation, days	13.0 (7.0–23.0)	15.0 (8.0–24.0)	14.0 (9.0–22.8)	0.206
In survivors at day 28, days	14.0 (8.0–27.0)	16.0 (10.0–30.0)	16.0 (10.0–26.0)	0.192
Tracheostomy—no (%)	47/241 (19.5)	86/527 (16.3)	53/322 (16.5)	0.517
Thromboembolic complications—no (%)	80 (32.8)	146 (27.5)	88 (27.2)	0.258
Pulmonary embolism	64 (26.2)	112 (21.1)	70 (21.6)	0.263
Deep vein thrombosis	12 (4.9)	29 (5.5)	14 (4.3)	0.776
Ischemic stroke	9 (3.7)	15 (2.8)	6 (1.9)	0.401
Myocardial infarction	5 (2.0)	9 (1.7)	2 (0.6)	0.261
Systemic arterial embolism	2 (0.8)	1 (0.2)	1 (0.3)	0.349
Acute kidney injury—no (%)	104 (42.6)	235/529 (44.4)	149/322 (46.3)	0.689
Need for RRT—no (%)	40 (16.4)	103 (19.4)	58 (17.9)	0.610
Need of rescue therapy—no (%) *	168/241 (69.7)	393/526 (74.7)	245/321 (76.3)	0.191
Prone positioning	124/241 (51.5)	300/527 (56.9)	188/322 (58.4)	0.232
Recruitment maneuver	9/200 (4.5)	36/440 (8.2)	18/271 (6.6)	0.239
Use of NMBA	95 (38.9)	266 (50.1)	166 (51.2)	0.006
ECMO	1/241 (0.4)	6/525 (1.1)	5/318 (1.6)	0.415
ICU length of stay, days	14.0 (8.0–26.0)	16.0 (9.0–26.0)	15.0 (10.0–26.0)	0.302
In survivors, days	15.5 (9.3–29.0)	18.0 (11.0–31.0)	17.0 (11.0–28.0)	0.379
Hospital length of stay, days	23.0 (13.0–37.3)	23.0 (13.0–36.0)	25.0 (16.8–37.0)	0.219
In survivors, days	29.0 (18.0–44.5)	30.0 (20.0–45.8)	29.0 (21.0–42.0)	0.682
ICU mortality—no (%)	76/238 (31.9)	186/519 (35.8)	85/313 (27.2)	0.034
Hospital mortality—no (%)	80/224 (35.7)	191/490 (39.0)	87/290 (30.0)	0.041
90-day mortality—no (%)	82/219 (37.4)	201/492 (40.9)	91/285 (31.9)	0.046

Data are median (first quartile–third quartile) or No (%). Percentages may not total 100 because of rounding. RRT: renal replacement therapy; NMBA: neuromuscular blocking agent; ECMO: extracorporeal membrane oxygenation; ICU: intensive care unit. * assessed in the first four days of ventilation.

**Table 3 jcm-10-01176-t003:** Adjusted effect of body mass index categories in 28-day, ICU, hospital, and 90-day mortality *.

	Effect Estimate (95% CI)	*p * Value
**28-day mortality**		
Body mass index category		
Normal	1 (Reference)	
Overweight	1.10 (0.83 to 1.47)	0.500
Obese	0.89 (0.63 to 1.25)	0.510
**ICU mortality**		
Body mass index category		
Normal	1 (Reference)	
Overweight	OR, 1.39 (0.96 to 2.00)	0.079
Obese	OR, 1.07 (0.69 to 1.65)	0.753
**Hospital mortality**		
Body mass index category		
Normal	1 (Reference)	
Overweight	OR, 1.38 (0.95 to 2.00)	0.090
Obese	OR, 1.05 (0.68 to 1.64)	0.817
**90-day mortality**		
Body mass index category		
Normal	1 (Reference)	
Overweight	HR, 1.16 (0.89 to 1.52)	0.270
Obese	HR, 1.00 (0.72 to 1.38)	0.999

Continuous variables were included after standardization, and the hazard ratio represents the increase in one standard deviation of the variable. OR: odds ratio; HR: hazard ratio. * All models adjusted for age, hypertension, heart failure, diabetes, chronic obstructive pulmonary disease, use of angiotensin converting enzyme inhibitor, heart rate, mean arterial pressure, pH, PEEP, fluid balance, use of vasopressor, and prone positioning on the day of start of ventilation.

## Data Availability

All data are available upon request.

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
