# Peer review of "Associations of Body Mass Index with Ventilation Management and Clinical Outcomes in Invasively Ventilated Patients with ARDS Related to COVID-19—Insights from the PRoVENT-COVID Study"

_jcm, 2021, doi:10.3390/jcm10061176_

Round 1

Reviewer 1 Report

The authors make a good analysis of the results of their practice in the ventilation of patients with ARDS - COVID 19 related (with more patiens and information than the first PRo-VENT-Covid publication) - focusing on the aspect of the obesity risk factor.

They also define an intermediate "overweight" population that is not normally mentioned in publications. Therefore it is also difficult to compare with other studies that choose the obese (> 30 Kg / m2) to describe differences.

However, I would suggest that you incorporate, in the interpretation of your results, the knowledge of two reviews on ventilation in obese patients with ARDS that are important to consider in this group of patients (regardless of COVID 19)

Jong et al. Critical Care (2019) 23:74
https://doi.org/10.1186/s13054-019-2374-0

Intensive Care Med (2020) 46:2423–2435
https://doi.org/10.1007/s00134-020-06286-x

The results (pg 8 line 177-181) talk about lower complaince of the respiratory system in obese patients "with higher mechanical power and needed higher oxygen fractions".

One of the limitations of the study is to consider only pulmonary compliance (driving pressure) as an indicator of total respiratory system compliance.

The relative part of pressure due to transthoracic pressure is often higher in the patient with obesity than in the patient without obesity
(elevated pleural pressure, which can be estimated by esophageal pressure). The plateau pressure represents the pressure used to distend the chest
wall plus lungs. In patients with obesity, elevated plateau pressure may be related to an elevated transthoracic pressure, and not an increase in
transpulmonary pressure with lung overdistension. 

These pathophysiological considerations maybe explain the results expressed in fig 3 where there is no significant difference in mortality in severe ARDS (between obese and normal weight) but even shows an important difference, in favor of obese patients in mild and moderate ARDS.

In the vast majority of centers, we still do not routinely measure esophageal pressure in these patients to adjust ventilation parameters with knowledge of transpulmonary and transthoracic pressure. As you actually comment, the compliance with existing guidelines to use limitation in volume tidal  "protective" was  better in COVID–19 patients, "because care for a surge of COVID–19 patients had to be provided by hospital personnel who had less experience or confidence with setting a ventilator and thus following the local guidelines more strictly".

Table 3 give show results difficult to interpret for the "overweight" group... (adjusted for other risk factors). Could you comment?

In the conclusions you may be able to reformulate, based on your results in a correct way.
"Lung protective ventilation and prone is feasible in all patients" is not new
"BMI should not be used in decision to proceed ventilation" would the opposite be ethically correct?

Congratulations on this collaborative research effort by the Dutch UCI's.

Author Response

  1. The authors make a good analysis of the results of their practice in the ventilation of patients with COVID–19 related ARDS (with more patients and information than the first PRoVENT–COVID publication) – focusing on the aspect of the obesity risk factor.

Thank you for your kind words. You’re very correct that we report here on more patients than in the first report on the PRoVENT–COVID study––that first report was an ‘urgent’ publication, reporting only on the first month of the outbreak in the Netherlands; the whole study, though, included 1122 fully analyzable invasively ventilated patients in the first three months of the national outbreak, and it is this final dataset that we used for the current analysis.

Patient flow is shown in Figure 1 (Line 174).

  1. They also define an intermediate ‘overweight’ population that is not normally mentioned in publications. Therefore, it is also difficult to compare with other studies that choose the obese (> 30 Kg/m2) to describe differences.

Correct, we chose to describe three groups, normal–weight, overweight and obese patients––in line with the predefined analysis plan for this analysis. We believe the extra information (by also reporting on overweight patients) adds to the current knowledge.

Since we report all data for all three groups in all parts of the analysis, we do think the data reported here can be compared to data reported on obese vs non–obese patients.

  1. I would suggest that you incorporate, in the interpretation of your results, the knowledge of two reviews on ventilation in obese patients with ARDS that are important to consider in this group of patients (regardless of COVID–19): Critical Care (2019) 23:74 (https://doi.org/10.1186/s13054-019-2374-0) and Intensive Care Med (2020) 46:2423–2435 (https://doi.org/10.1007/s00134-020-06286-x)

Thank you for these citations. We actually used these 2 seminal papers in preparing the manuscript, in particular for identifying relevant references for the current report. In the revised version we now also refer to these 2 papers, as follows:

Esophagus catheters were seldom used in this cohort of patients, probably because hospital personnel with little experience with setting a ventilator also had less knowledge of transpulmonary and transthoracic pressure measurements to adjust ventilator settings. It remains uncertain whether (adequate) use of esophagus pressure measurements would have altered the current findings […].’ (line 276-280)

Both strategies have been proven to be very effective in patients with ARDS […, …] and have been advised in obese patients with ARDS from another origin […] …’ (line 346-347)

  1. The results (Page 8 of 16; line 177-181) talk about lower compliance of the respiratory system in obese patients ‘with higher mechanical power and needed higher oxygen fraction’. One of the limitations of the study is to consider only pulmonary compliance (driving pressure) as an indicator of total respiratory system compliance. The relative part of pressure due to transthoracic pressure is often higher in the patient with obesity than in the patient without obesity (elevated pleural pressure, which can be estimated by esophageal pressure). The plateau pressure represents the pressure used to distend the chest wall plus lungs. In patients with obesity, elevated plateau pressure may be related to an elevated transthoracic pressure, and not an increase in transpulmonary pressure with lung overdistension. These pathophysiological considerations maybe explain the results expressed in Figure 3 where there is no significant difference in mortality in severe ARDS (between obese and normal weight) but even shows an important difference, in favor of obese patients in mild and moderate ARDS.

Correct. This is an important limitation of the current analysis, and this is now discussed, in detail, in an extra paragraph in the Discussion. This paragraph reads as follows:

One important limitation of this analysis that we could not use esophageal pressures to calculate pulmonary compliance––indeed, we could only report total respiratory system compliance. In an obese patient, an elevated plateau pressure may be related to an elevated transthoracic pressure, and not necessarily an increase in transpulmonary pressure. This may explain, at least in part the finding that there is no significant difference in mortality in severe ARDS (between obese and normal–weight), and even the important difference in favor of obese patients in mild and moderate ARDS.’ (line 308-323)

  1. In the vast majority of centers, we still do not routinely measure esophageal pressure in these patients to adjust ventilation parameters with knowledge of transpulmonary and transthoracic pressure. As you actually comment, the compliance with existing guidelines to use limitation in volume tidal "protective" was better in COVID–19 patients, ‘because care for a surge of COVID–19 patients had to be provided by hospital personnel who had less experience or confidence with setting a ventilator and thus following the local guidelines more strictly’.

Yes, and this may explain why in most if not all cases esophagus pressure measurements were not performed, and if so whether the measurements were used adequately – we comment on this is in Discussion as follows:

Esophagus catheters were seldom used in this cohort of patients, probably because hospital personnel with little experience with setting a ventilator also had less knowledge of transpulmonary and transthoracic pressure measurements to adjust ventilator settings. It remains uncertain whether adequate use of esophagus pressure measurements would have altered the current findings.’ (line 276-280)

  1. Table 3 give show results difficult to interpret for the ‘overweight’ group... (adjusted for other risk factors). Could you comment?

Thank you. In the Results the following is written:

Among secondary outcomes, ICU–, hospital–, and 90–day mortality was lower in obese patients in the unadjusted analysis (Table 2 and Figure S5), but not in the adjusted analysis (Table 3).’ (line 215-217)

  1. In the conclusions you may be able to reformulate, based on your results in a correct way. ‘Lung protective ventilation and prone is feasible in all patients’ is not new ‘BMI should not be used in decision to proceed ventilation’ would the opposite be ethically correct?

Thank you for this comment. We changed this into:

‘Comparable to normal–weight, overweight and obese patients with ARDS from another origin, lung–protective ventilation with a lower tidal volume and prone positioning is very feasible in normal–weight, overweight and obese patients with ARDS related to COVID–19.’ (line 353-356)

We agree, in our eyes it is NEVER ethical to base such important decisions on BMI. However, in the context of surges of patients and shortages of ICU beds and ventilators, this approach has been suggested, and actually we see colleagues suggesting BMI predicts poor outcomes – this is not true, neither in patients with ARDS from another origin than COVID–19, nor patients with COVID–19 ARDS. This was one of the reason for why we performed the current analysis.

The last lines of the Conclusion now read as follows:

Obese COVID–19 patients with ARDS do not have a worse outcome compared to normal–weight and overweight patients––therefore, obesity should not be considered in the decision who gets or will continue with intubation and invasive ventilation.’ (line 356-358)

  1. Congratulations on this collaborative research effort by the Dutch ICUs.

Thank you, and thank you for your helpful comments.

Reviewer 2 Report

The authors present a secondary analysis of a retrospective multi-center study on influence of BMI on clinical practice and outcome in mechanically ventilated COVID19 patients. The paper is well written and covers a relevant clinical topic. I have just a view comments.

Major comments

  • The time period used for analysis was only four days what is a major limitation of this study. The authors need to justify this number of days. Was this period chosen based on the available data or was this defined previously in the protocol of the initial study?

Minor comments

  • Table 1 , mechanical power: there are different formulars available to calculate mechanical power. The authors should give the formula which was used. Mechanical power was statistically higher in obese patients. This should be discussed.What was the reason? Higher PEEP, higher VT, or both?
  • Table 2: in this table 28-day mortality is given as the primary outcome parameter of this study. This is in contrast to the outcomes given in the material section.
  • Figure 3: survival is better in obese patients with mild ARDS. This effect is gone after adjustments. This need to be discussed. Is it possible that a mild ARDS is relatively overdiagnosed due to derecruitment which causes oxygenation impairment rather than due to real ARDS?
  • Lines 217-220: I do not understand the difference between these results and the results given in lines 171-176
  • Lines 221-222: at all it is a bit confusing that results are analyzed before and after adjustments and overall and in several obesity subclasses. What is the meaning of the finding which is presented in this paragraph?
  • Line 275. There is a dot after reference [24]
  • Line 280: Braces are missing for reference #13

Author Response

  1. The authors present a secondary analysis of a retrospective multicenter study on influence of BMI on clinical practice and outcome in mechanically ventilated COVID19 patients. The paper is well written and covers a relevant clinical topic. I have just a view comment.

Thank you for your kind words and the comments.

  1. The time period used for analysis was only four days what is a major limitation of this study. The authors need to justify this number of days. Was this period chosen based on the available data or was this defined previously in the protocol of the initial study?

This was predefined in the study protocol. We agree that 4 days is relatively short––however, we chose to collect ventilation data with high granularity, and ‘the collection of ventilation variables and adjunctive treatments was restricted to the first 4 calendar days of ventilation to keep the workload of the study at an acceptable level.’ This was explained in Lancet Respir Med 2021; 9(2):139-148, the first report on this study. To prevent repetition, we choose not to add this information to the current text––however, if you feel this is necessary we can adjust the manuscript, of course

  1. Table 1. Mechanical power: there are different formulas available to calculate mechanical power. The authors should give the formula which was used. Mechanical power was statistically higher in obese patients. This should be discussed. What was the reason? Higher PEEP, higher VT, or both?

Apologies for not reporting how we calculated mechanical power. This is now added to the manuscript, in the Methods as follows:

The driving pressure and mechanical power of ventilation were calculated as follows––driving pressure (in cm H2O) = peak pressure – positive end–expiratory pressure (PEEP); and mechanical power (in J/min) = 0.098 * tidal volume * respiratory rate * (peak pressure – 0.5 * driving pressure).’ (line 101-104)

We also added the following to the Discussion:

It may also explain the finding that obese patients received ventilation with a higher driving pressure and higher mechanical power, and had a lower respiratory system compliance.’ (line 321-323)

  1. Table 2: in this table 28–day mortality is given as the primary outcome parameter of this study. This is in contrast to the outcomes given in the material section.

Apologies for the confusion caused by reporting this in this way. You’re right that a combination of key ventilator settings was the primary outcome. 28–day mortality was the ‘primary’ clinical outcome. We deleted ‘Primary outcome’ and ‘Secondary outcomes’ from Table 2.

  1. Figure 3: survival is better in obese patients with mild ARDS. This effect is gone after adjustments. This need to be discussed. Is it possible that a mild ARDS is relatively over–diagnosed due to derecruitment which causes oxygenation impairment rather than due to real ARDS?

Correct suggestion. This is one of the reasons for why survival benefit was not found after adjustments. We added this to the Discussion, as follows:

While survival seemed better in obese patients with mild ARDS, this finding disappeared after adjustments. It is possible that mild ARDS was relatively over–diagnosed due to derecruitment, causing oxygenation impairments rather than that a patient really had ARDS.’ (line 255-259)

  1. Lines 217-220: I do not understand the difference between these results and the results given in lines 171-176

Apologies – we do not understand what you’re referring to: in the file formatted by the journal the first part (lines 217-220) is about the adjusted analysis, and the second part (lines 171-176) deals with ventilator settings; in the file submitted by us the first part (lines 217-220) describes the cohort, and the second part (lines 171-176) discusses part of the analysis plan. We are afraid there is a mix–up of lines?

  1. Lines 221-222: at all it is a bit confusing that results are analyzed before and after adjustments and overall and in several obesity subclasses. What is the meaning of the finding which is presented in this paragraph?

This part is a sensitivity analysis, to see if difference remain after adjustments in obesity subclasses.

  1. Line 275. There is a dot after reference [24]

Corrected into a comma.

  1. Line 280: Braces are missing for reference #13

Corrected, braces are added.